# A cascading model for nudging employees towards energy-efficient behaviour in tertiary buildings

Ilias Kalamaras[1], Rubén Sánchez-Corcuera[2], Diego Casado-Mansilla[3]*, Apostolos C. Tsolakis[1], Oihane Gómez-Carmona[2‡], Stelios Krinidis[1‡], Cruz E. Borges[2‡], Dimitrios Tzovaras[1‡], Diego López-de-Ipiña[3‡]

**1** Information Technologies Institute - Centre for Research and Technology Hellas, Thessaloniki, Greece, **2** DeustoTech, University of Deusto, Bilbao, Spain, **3** Faculty of Engineering, University of Deusto, Bilbao, Spain

☯ These authors contributed equally to this work.
‡ OGC, SK, CEB, DT and DL also contributed equally to this work.
* dcasado@deusto.es

**Data Availability Statement:** The data is on Zenodo or upload to this journal repository: -

## Abstract

Energy-related occupant behaviour in the built environment is considered crucial when aiming towards Energy Efficiency (EE), especially given the notion that people are most often unaware and disengaged regarding the impacts of energy-consuming habits. In order to affect such energy-related behaviour, various approaches have been employed, being the most common the provision of recommendations towards more energy-efficient actions. In this work, the authors extend prior research findings in an effort to automatically identify the optimal Persuasion Strategy (PS), out of ten pre-selected by experts, tailored to a user (i.e., the context to trigger a message, allocate a task or providing cues to enact an action). This process aims to successfully influence the employees' decisions about EE in tertiary buildings. The framework presented in this study utilizes cultural traits and socio-economic information. It is based on one of the largest survey datasets on this subject, comprising responses from 743 users collected through an online survey in four countries across Europe (Spain, Greece, Austria and the UK). The resulting framework was designed as a cascade of sequential data-driven prediction models. The first step employs a particular case of matrix factorisation to rank the ten PP in terms of preference for each user, followed by a random forest regression model that uses these rankings as a filtering step to compute scores for each PP and conclude with the best selection for each user. An *ex-post* assessment of the individual steps and the combined ensemble revealed increased accuracy over baseline non-personalised methods. Furthermore, the analysis also sheds light on important user characteristics to take into account for future interventions related to EE and the most effective persuasion strategies to adopt based on user data. Discussion and implications of the reported results are provided in the text regarding the flourishing field of personalisation to motivate pro-environmental behaviour change in tertiary buildings.

https://zenodo.org/records/2610102 - https://zenodo.org/records/3565757 -https://doi.org/10.5281/zenodo.10377229 - https://github.com/morelab/st_recommender.

**Funding:** Ministry of Economy, Industry and Competitiveness of Spain for IoP, under Grant No.: PID2020-119682RB-I00.

**Competing interests:** NO authors have competing interests.

## Introduction

Global contribution from buildings in energy consumption, both residential and non-residential, has steadily increased, reaching levels of 40% in Europe, surpassing the other significant sectors of industry and transportation [1]. At the same time, over the years, occupant behaviour has been vastly recognised as a crucial factor for Energy Efficiency (EE) [2, 3]. For this reason, together with the advent of the integration of Internet of Energy (IoE) innovations in residential buildings [4], extensive research has been invested into creating recommendation systems that can infer the optimal recommendation to a user in terms of EE, taking into account the user preferences and comfort levels [5, 6]. Furthermore, it has been identified that buildings occupied by users with wasteful energy behaviour can have twice the consumption as the ones that energy savers generally occupy [7]. This pattern of consumption behaviour is especially pronounced in developing and transition economies [8]. This highlights the fact that awareness and energy-related behaviour hold vast potential towards EE. This occupant energy-related behaviour potential can reach up to 30% for tertiary buildings (infrastructure occupied by public authorities, associations and companies providing services) [9]. For households on the other hand, feedback-based approaches, such as those employing edge-based IoE platforms for energy data analysis and behavioural change [10], can reduce energy consumption up to a realistic 5 to 10%, and only under specific conditions, according to an extended study from JRC [11]. Thus, researchers are exploring ways to affect the daily behaviour of individuals so that this significant energy-saving potential can be exploited in the short- and long-term [12–14]. Moreover, comprehending the nuances of waste management behaviour—which includes the reduction, reuse, recycling, and proper disposal of waste as influenced by rational benefit-cost analyses, social pressures, and personal environmental beliefs and attitudes— plays a pivotal role in enhancing energy efficiency [15], together with detecting anomalous energy consumption [16] and selecting appropriate technologies based on climatic conditions and specific building [17].

There are quite a few ways to affect user behaviour. Energy-saving recommendation systems are proven to be a potential approach for promoting building sustainability and energy efficiency [5]. Still, according to Himeur et al., future research should focus on improving the quality and applicability of recommendation frameworks. The use of micro-moment recommendations for reshaping consumption habits, as evidenced in a case study [18], is one such area of improvement. In this line, one particular challenge in this respect is how to persuade, or nudge, the end-user in following the energy-saving recommendations or advice. Computational persuasion, being based on computational models of argument, has been identified as one of the most promising technological approaches for behaviour change applications [19]. According to Jesse and Jannach, who examined the relationship between digital nudging and recommender systems, through the automatic selection of the offered content, these systems control which information is easily accessible to us and hence influence our decision-making processes [20]. In this sense, the emergence of explainable intelligent systems [21] and the integration of sensor data and human feedback for personalized energy-saving recommendations [22] becomes crucial and further contributes to this process by delivering explainable and personalized recommendations for energy efficiency.

Based on this review, it can be observed that there is a variety of persuasion strategies that one can follow to promote a behaviour, e.g. providing rewards, selecting a bespoke activity to an end-user or exploiting social recognition; however, different end-users may be influenced to a different degree by the same persuasion strategy. Therefore, it is important to select a personalised persuasion strategy for each individual to have a higher impact in the long term.

The work presented is also motivated by the success of personalized recommendation systems, as implemented by popular platforms, such as Netflix, Amazon, Booking or IMDB, which recommend particular items in a personalized manner, thereby increasing the platform acceptance rate and eventually altering the behaviour of end-users. Such systems, instead of providing a single best choice, offer a set of choices, ranking them according to their appropriateness for an individual. To provide the proper recommendation at the appropriate time, the history of user choices is being used, along with user preferences and personal profiles [23]. If such information is not available, the so-called Cold-Start Problem (CSP) arises. To alleviate this problem, the recommender systems might employ user profiles [24], short surveys [25] or federated learning for generating energy efficiency recommendations [26].

The current work follows similar principles, but in a different scope: instead of dealing with the actual message, the item in typical recommender systems, to deliver to the end users to enhance their energy-saving behaviour, the manuscript deals with the best persuasion strategy to use to approach a user, i.e. how to present the message (i.e., the context, the channel or format), so as to persuade the user. In this sense, the recommendation of the context is not directly towards the end-user, but rather towards an information system installed in a building that recommends or assigns actions to do to employees in favour of EE. Such kinds of systems can be promoted by building managers, energy managers or building planners. Therefore, the type of message to be delivered or its content is out of the scope of the presented research and should be provided by those managers who understand the energy necessities of each work environment. To give an example, if the system proposed was the well-known music platform Spotify and the company was trying to make a user sign up to the premium version, our recommendation framework would suggest to Spotify the best channel to do so or the most convincing strategy to present the message in order to influence the listener (e.g., using the social media profiles of the user or telling to him/her who many friends and peers are already premium), yet not the message's content itself.

Considering this previous information, the contribution of this work is the following. It proposes a machine learning-based method for identifying the most appropriate persuasion strategy for an individual, for delivering more tailored recommendations towards altering their behaviour into a more energy-efficient one. Therefore, the presented manuscript does not aim to deliver a recommender system for recommending items or actions to perform (i.e., the core content), but rather to support recommender systems to select persuasion strategies to follow to more effectively deliver a tailored to the specific individual recommendation (i.e. the context). To the best of our knowledge, such data-driven personalised approaches for persuasion have not been explored much in the literature, with the current work being the first AI-based framework that decides which is the best way to present a motivational message to users based on explicit profiling through an ensemble of methods (i.e., cascading).

Although this work is based on earlier work by the authors [14, 25] where it was found that AI-based methods were able to outperform statistical approaches, it goes a step further over these works and fuses them in a two-step methodology, thus increasing the accuracy of the strategy predictions. First, a predefined set of persuasion strategies is ranked per user according to their self-reported individual profile, such as socio-economic information, types of behaviour and work habits, by extending previously employed online survey results from tertiary buildings and making use of matrix factorization methods. Subsequently, this ranking is used as a filtering step to a random forest regression model used to identify the optimal strategy tailored to each individual (sequential model fusion, which can be considered as a field within the ensemble paradigm [27]). To the authors' knowledge, this approach is the first of its kind and has not been applied to any other domain, including the energy one.

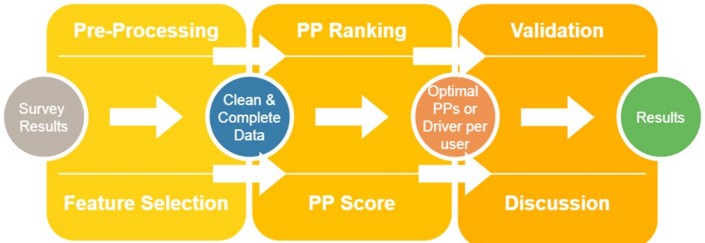

**Fig 1. Adaptive dual recommendation framework methodology is divided into three main steps: Data gathering, persuasion principle recommendation and validation.**

The manuscript is organised as follows: In Section Materials and Methods, the overall methodology designed and followed is presented, including the dataset description and the detailed implementation of the two models and their fusion. The experimental results are documented in the next section followed by an extensive discussion including the limitations of the conducted research. Finally, the manuscript is closed with a Results section that provides concluding remarks.

## Materials and methods

In this section, the methodology followed for carrying out the experimentation detailed in this article is presented in detail. Fig 1 shows a schematic of the workflow. It starts with collecting data through surveys, and its subsequent cleaning and selection of the most representative characteristics employing feature selection methods. Afterwards, the proposed persuasion principle recommendation methods are applied to select the most suitable persuasion principles for the users. The recommendation results are validated and discussed in the next sections of this manuscript.

Initial versions for the proposed recommendation methodology and the data used have been based on the GreenSoul H2020 Research and Innovation Action project [28], which used ICT-based interventions to alter the energy-related behaviour of end-users, following several recommendation principles/strategies. Therefore, before describing the hereby proposed recommendation methods in detail, some context will be provided in the following sections about the information used as the foundation for this work, i.e. the survey design, dataset collection and persuasion principle definition. Most of this background information has been detailed in [14, 29], so in this paper, only an overview is presented for the sake of understanding the presented methodology. Through the GreenSoul project's activities, a detailed analysis of the human factors that affect energy efficiency-related decisions was performed, leading to various behavioural models that can be used as core components of the ICT interventions for raising awareness and affecting user actions. One of the outcomes of this approach was designing a survey for adequately identifying the persuasion principles that are most appropriate per user towards achieving the utmost in terms of energy efficiency.

### Survey design

In order to infer the preferences of the end-users and alleviate the previously mentioned Cold-Start Problem (CSP), a socio-economic survey was conducted in 7 tertiary buildings across Europe [28]. The survey took place in two phases, before and after the pilot intervention. The survey structure was slightly altered in the second phase (we removed some questions) based on the first phase results. The surveys included questions regarding demographics and work

and non-work-related energy efficiency, e.g. confidence in the use of technology, potential barriers for energy-saving behaviour or intentions to join pro-environment actions, resulting in 9 predictor variables (The full survey can be consulted in S1 File). For a full immersion on the data collected in both surveys, these are publicly available on Zenodo; namely, the pre-pilot dataset [30], with 303 responses, and the post-pilot [31] dataset, with 65 responses. However, as 368 samples were insufficient, we decided to extend the evidence by running a second iteration of the survey through the Prolific platform (https://www.prolific.co/). The new data collected presents an additional 375 responses from the same four countries. Furthermore, we used a similar screening methodology employed in the first survey during the GreenSoul project. By doing so, the analysis was expected to have more objective results as the two datasets are entirely independent and with around two years of difference. The final dataset with N = 743 with all the answers already coded can be found also in Zenodo platform [32].

## Datasets

The two datasets used in this that were merged are:

- the GreenSoul dataset with data from 368 users, and

- the Prolific dataset, with data from 375 users beyond the context of the EU project.

The records in each dataset consist of ten "Persuasion features" (see next section), i.e. the self-reported answers to questions about the appropriateness of the persuasion principles, used as the target variables in the developed models. Besides, nine "User features", i.e. demographic and energy-related behaviour questions used as the predictor variables. Table 1 offers a listing of the persuasive principles included in this study, along with a brief explanation. For the developed models, these are the values that the "principle" categorical variable takes. The user features can be also consulted in Table 2.

**Persuasion principles.** In the aforementioned surveys, ten persuasion principles (PP) have been used as targets or persuasion features. All ten principles were selected from a larger set of principles composed through an extensive survey of the literature [33–37]. Specifically, as also mentioned in [38], the persuasion strategies collected from the above literature were grouped into 15 initial common persuasion principles that correspond to general modes of thinking that affect people to be persuaded, as identified by the literature experts. The survey participants were asked to rank these principles according to their perceived applicability to

**Table 1. Persuasive principles/strategies.**

| Persuasive Principle | Description/Examples | Type of Interaction |
|---|---|---|
| Authority | The system is an expert on energy efficiency. | People |
| Cause and effect | Visualisation of the outcomes if the desired action is achieved. | System |
| Conditioning | Provide incentives/rewards for certain actions. | People |
| Physical | Create digital or physical interfaces with | System |
| attractiveness | aesthetics in mind. | |
| Reciprocity | Give hints about the efficiency gained by the system. | System |
| Self-monitoring | Provide tools where the users can see their consumption | System |
| Similarity | Find peers that can give advises to target users through social networks or platform. | People |
| Social proof | Show the number of followers of the system. | People |
| Social Recognition | Showing in public that someone is the best of the month. | People |
| Suggestion | Provide hints/cues just-in time or about-to moments. | System |

**Table 2. The nine most important features extracted from the dataset and their descriptions.**

| Name | Description | Scale/values |
|---|---|---|
| Age | Age group of the user. | <21, 22–40, 41–52, 53–71, >72 |
| Barriers | Assess if the user has any special barrier to behave pro-environmentally in the workplace. | Absentminded, uncertain, discouraged, none |
| Confidence | Assess end-user confidence in the technology. | Confident, not confident |
| Country | Country of the user. | Spain, Greece, Austria, UK |
| Education | Education level of the user. | None, secondary, post-secondary, university, post-graduate, doctoral |
| Gender | Gender of the user. | Female, male, other |
| Intentions | Assess if the degree a user is willing to behave pro-environmentally. | Pre-contemplation, contemplation, action |
| Work culture | The work culture of the end-user's job (e.g. competitive, cooperative, etc.). | Teamwork, goal-oriented, encourage creativity, formal and hierarchical, none of them |
| PST | User archetype | Pinball, shortcut, thoughtful [44] and their combinations |

them. The ten PPs that are selected for the current study were the most influential for the end-users out of the original 15, based on their applicability, and were used to design and evaluate the proposed methods. Furthermore, the final ten principles were validated by two experts with a strong background in persuasion and nudge theory. The selected PP are the following: 1) Authority, 2) Cause and Effect, 3) Conditioning, 4) Physical Attractiveness, 5) Reciprocity, 6) Self-monitoring, 7) Similarity, 8) Social Proof, 9) Social Recognition, and 10) Suggestion. These PPs are also grouped into two clusters: 1) based on the interaction with a system, or 2) based on the interaction with other people. The objective is to examine the main driver behind different types of user profiles. Thus, to understand if people are more oriented to 1) a PP that requires more cognition and interaction with the system (central route of persuasion [39]); 2) or if the people prefer to be more influenced by peers as social actors (normative behaviour [40]). It is worth mentioning that the statistical analysis of the top 10 preferred principles by all the users in the full dataset was: "Conditioning", "Social-recognition", "Physical attractiveness", "Self-monitoring", "Reciprocity", "Social-proof", "Cause and Effect", "Similarity", "Suggestion" and "Authority". See Fig 2.

## Pre-processing & feature selection

The input data features consist of the answers of the users to the energy-related questionnaire. Each row of the data corresponds to a user and each column to a questionnaire answer. The data are a mixture of ordinal and categorical variables, which are handled differently. Ordinal variables, such as age and education level (on a scale from 1 to 5), are directly transformed to numerical. Variables with different ranges for the scales, such as rankings from 1 to 10, are linearly transformed to the 1–5 scale.

A series of initial transformations were made to limit the number of original features (37) to a smaller number of features that contain most of the information for the users and are few enough to be easier to collect in future endeavours. The first transformation was performed towards ensuring inter-rater reliability. The features that provide the least amount of information about the users and require more effort from users to answer were removed. With this initial transformation, the user feature vector's dimensionality was reduced from 37 features to 21.

Subsequently, feature selection algorithms were applied to detect the importance of each user feature in the classification and reduce the dataset's dimension. This transformation's objective was to identify the minimum subset of information that would represent a user to

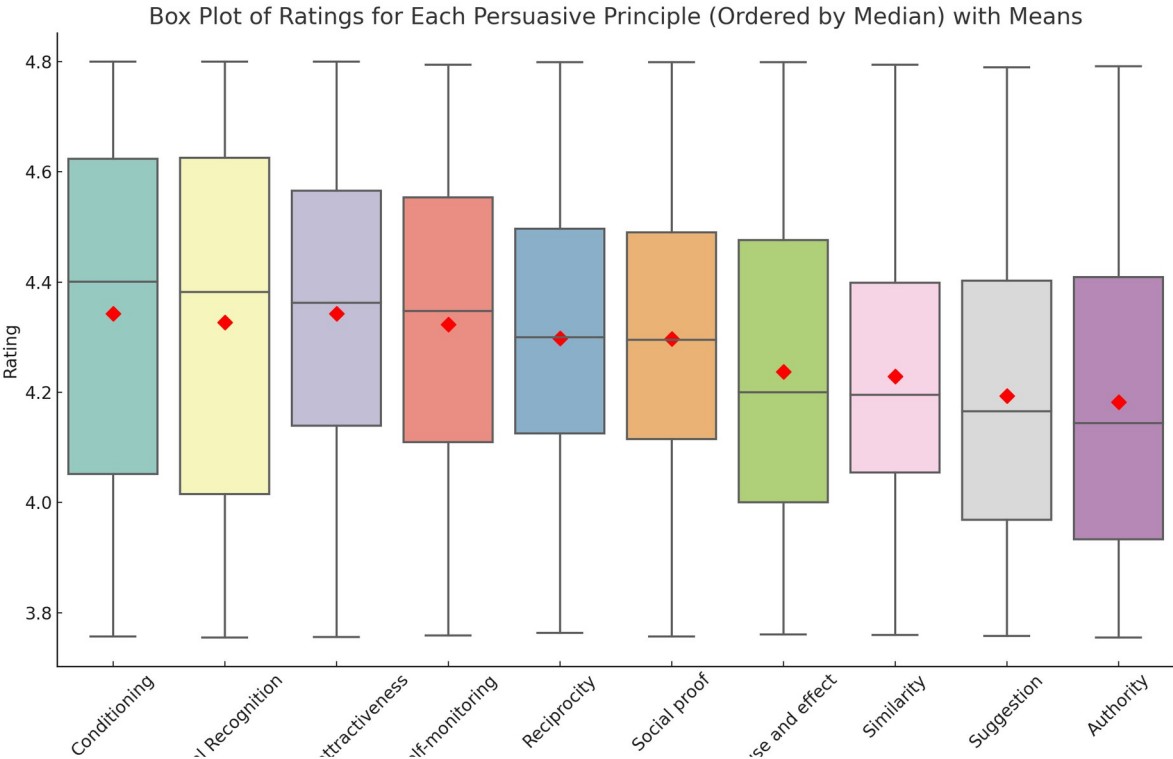

**Fig 2. A boxplot of the top-rated persuasive principles in the dataset.** The medians are represented with diamonds. These PPs are represented in descending order.

generate the best persuasive principle for each user while balancing the size of the dataset and the information loss.

Some details about the feature extraction methods used are as follows. Decision trees [41] were used as feature selection algorithms for this experiment as they have proven to be suitable for this task [42]. Two different classifiers, Random Forest and Extra Trees, were employed 10.000 epochs and 100 estimators, results were averaged and compared to decide which user-input features provide the most information for selecting the top principle of persuasion (i.e., a classification task). After this transformation, the user feature vector was reduced to the nine (9) most important user features shown in Table 2. We decided to maintain the features that provide 80% of the information regarding the users in the dataset as stated in [44]. This dimensionality reduction process also led to the Prolific survey's design, which included only these nine most crucial user features as questions.

As a result of the feature selection process, nine predictor variables have been extracted, consisting of the user answers regarding demographics and behavioural aspects as can be observed in the final survey design in S1 File. These variables will be used as predictor variables by the models to predict the rankings and scores of the persuasion principles.

The ranks produced by the first model of the ensemble are also used as scores, by linearly transforming them to the 1–5 scale. The ensemble's goal is to use the user profile, built from demographics and energy-related behaviour at work, to predict the rankings and scores and thus suggest the most appropriate persuasion principle for a user.

## Proposed method

Based on the structure of these datasets, two different approaches, originally developed by the authors using preliminary data in previous publications, have been followed to identify the appropriateness of each persuasion strategy per user, each resulting in its recommendations, subsequently, these two sets of recommendations are merged in a step-by-step fashion, resulting in aggregated results. The first method [14], called here "Persuasion Principle Optimal Ranking", is based on an extended version of matrix factorization (MF) [45] through Factor Machines based on the library LightFM [46]. Specifically, our approach extends traditional matrix factorization (MF) by incorporating latent factors that represent both users and items. For users, these latent factors are derived from socio-economic features, such as age, income level, and occupation, as specified in Table 2. For items, the latent factors correspond to the characteristics of the persuasive principles being recommended, like their complexity, type, and targeted behaviour change. The elements of the matrix to be factorized are the interactions between users and items, quantified based on the users' responses to different persuasive principles. This extended MF approach allows us to capture the nuances of user preferences and strategy effectiveness, leading to more personalized and effective recommendations. This model results in personalised rankings of the selected persuasion principles from the most appropriate for a user to the least appropriate. The second method [25], called here "Persuasion Principle Score Prediction", is based on Random Forest regression [47] and results in personalised predictions of the score that a user would give to a persuasion principle on a scale from one to five. The former model can predict the order in which a user prefers to receive a persuasive message (ranking), and the latter provides the impact that this persuasive message will have on the specific user through ratings.

This section briefly describes the methods in isolation [14, 25] for the sake of completeness and introduces the fusion scheme used to aggregate the results of the two models. It is also essential to state that the extended dataset is analysed in this manuscript, presenting additional results both for the individual and the fused methods. Various scenarios have been evaluated with the sequential fusion of the two models (https://github.com/morelab/st_recommender).

**Persuasion principle optimal ranking.** The first model of the ensemble provides optimal rankings of the best persuasion principles for every user employing user features. As introduced above, a particular case of MF technique [45] was employed for the development of the model due to its popularity in recommendation systems. The model uses the features from persuasive principles and meta-data from users and their interactions to calculate the affinity between them and create the best persuasive principles for each user. MF-based systems are commonly employed for developing recommender systems, and in addition to being known to model reality very accurately, they are very good at updating the popularity of items or the likes and dislikes of users based on their activity.

The proposed model is developed using the LightFM library [46] that allows us to create a hybrid latent representation model [48]. Through this library, a model that enhances the MF method can be created by providing a method to add features to either users and items, which can be constructed as a particular case of Factorization Machines [49]. Users and items may be represented by a vector, called latent vector or embedding, containing those features through this method. Furthermore, features from users and items are also described as a scalar bias that represents the bias for every user and item. To infer how a user will rate a specific item, we employ the bias and the latent vectors for that user and item. A more detailed description of the model is presented in [14, 38].

**Persuasion principle score prediction.** The second model uses Random Forest regression to predict a user's scores for each persuasion principle in a questionnaire. Information about

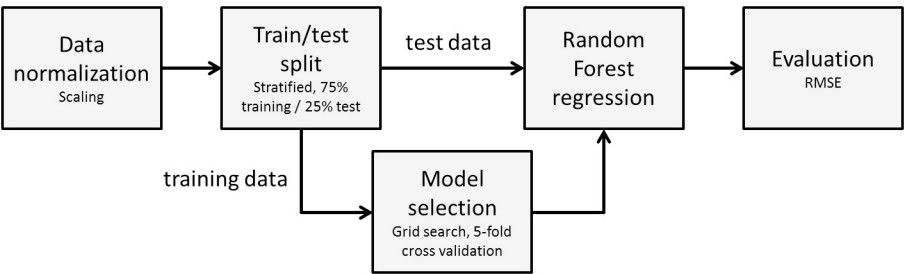

**Fig 3. Recommendation model workflow (Fig from [25]).**

the user demographics and her energy-related behaviour are used as predictors, while the scores of the users for the persuasion principles are used as the targets of regression. Details of the method have been presented in [25]; however, a brief description of the method is provided here for completeness. Fig 3 displays the overall workflow of the method.

The problem of predicting the persuasion principle scores is viewed as a regression problem. We aim to predict numerical values using a set of numerical and categorical predictors. We have used Random Forest (RF) regression [50, 51] for this task. Random Forests are ensembles of decision trees, each considering a different random subset of the input data and the input variables, while the final output is a majority voting over the decisions of all trees in the forest. Using multiple trees constructed from random subsets of the data leads to less overfitting and thus better accuracy when presented with novel samples. Random Forests are popular as non-linear prediction models, capturing the non-linear structure of the data and producing accurate predictions.

As a choice for regression, the Random Forest model is more complex than simpler alternatives, such as multi-variate regression, which can also be used to tackle such problems. However, it has been chosen due to its better prediction performance compared to other models. To demonstrate the better performance of this model over more classical alternatives, we have also considered a reduced rank multi-variate regression model (MVR). In MVR, all targets are modelled simultaneously, while the coefficient matrix is constrained to be of rank lower than the number of principles (10), to cope with any correlations between the targets. Apart from being a simpler alternative to Random Forest regression, MVR can also provide added insights into the importance of the predictors for predicting the outcomes.

There are multiple target variables in the current problem, one for each persuasion principle, so multiple RF models are trained. We build an RF model from the data for each persuasion principle, with the respective principle as the (scalar) target. When the input data of a new sample user are given, the multiple models will each produce a prediction of the score for the respective persuasion principle, which can then be used to suggest the most appropriate strategy.

To fit the RF models, the available data are split into training (75%) and testing (25%) sets using stratified sampling. The quality measure to determine the best split is variance reduction, with Mean Squared Error as the measure of variance. The number of trees in the forest is set to 1000.

**Fusion of optimal ranking and score prediction.** The two methods described above lead to two different types of results: the first method produces an optimal ranking, i.e. order, of the persuasion principles for a user, ordered from the most appropriate to the least appropriate one; the second method produces scores of appropriateness for each persuasion principle, for

a user. Apart from using these results individually, they have also been fused for increased performance.

To introduce some notation, let $r = (r_1, r_2, \ldots, r_m)$ be a ranking of the $m$ persuasion principles, with each $r_i$ taking values in the set $\{1, 2, \ldots, m\}$. A $r_i$ value of 1 means a high ranking, i.e. that the $i$-th principle has been ranked as the most appropriate, while a $r_i$ value of $m$ means a low ranking, i.e. that the $i$-th principle has been ranked as the least appropriate. Similarly, let $s = (s_1, s_2, \ldots, s_m)$ be a set of scores of the $m$ principles, with each $s_i$ taking values in the range $\{1, \ldots, 5\}$. A $s_i$ value of 1 means a low score, i.e. that the $i$-th principle is not preferred, while a $s_i$ value of 5 means a high score, i.e. that the $i$-th principle is preferred.

Fusion of the two types of output is performed sequentially. The ranking produced by the first method is used as a kind of filter for fine-tuning the results of the second method. Specifically, the ranking $r$ produced by the first method is used to keep only the top 5 highest ranked principles $I_5$:

$$I_5 = \{i \in \{1 \ldots m\} : r_i \leq 5\}. \tag{1}$$

Then, the scores of these top 5 principles, $S_5$, are taken from the score vector $s$ returned by the second method:

$$S_5 = \{s_i : i \in I_5\}. \tag{2}$$

Finally, the scores in $S_5$ are sorted to get the modified ranking $r'$:

$$r'_i = \text{order of } s_i \text{ in } S_5. \tag{3}$$

This modified ranking of 5 elements is the fused ranking and can be used, e.g. to extract the top 1 principle to compare with the top ground truth principle.

Hence, the fusion strategy in our model, combining different outputs sequentially, was chosen for its efficacy in integrating varied results: the optimal ranking and appropriateness scores for persuasion principles. The first method's ranking filters top principles, which the second method then refines with scores. This cascade approach, akin to dimensionality reduction simplifying the model's complexity while maintaining robust predictive power [52], aligns with previous methodologies and is well-suited for handling categorical variables directly. It leverages each model's strengths, sequentially refining the analysis for more precise, user-tailored recommendations.

Conceptually, fusing the optimal ranking estimation of the first step to the rating estimation of the second step is similar to the methodology followed in [14]. In that work, the second step was implemented using an Active Learning scheme employing Gaussian Process regression. Active Learning is most suitable in cases with limited learning samples, whereas in the current case, we have more data points available, allowing the use of batch methods such as Random Forest. Compared to Gaussian Processes, Random Forest has the added advantage that it can be applied directly to categorical variables and thus be used to compute a rating directly from the input data. This makes it independent from the output of the first step, which is beneficial for ensemble learning methods.

## Results

### Evaluation measures

**Root mean square error.** The Root Mean Square Error (RMSE) is used when the target variable is numerical, i.e., regression-type problems. If $x \in \mathbb{R}^n$ is the vector of ground truth

values of the target variable and $\hat{x} \in \mathbb{R}^n$ is the vector of predictions, the RMSE is defined as:

$$RMSE = \sqrt{\frac{1}{n}\sum_{i=1}^{n}(x_i - \hat{x}_i)^2}. \tag{4}$$

**F1-score.**   Also known as F-measure or F1, in most statistical findings this is defined as the harmonic mean of the *precision* and *recall*. It ranges from zero to one and can deal with cases where there is an uneven class distribution. In general, high values of F1-score indicate high classification performance. Since *F1-score* is calculated from *precision* and *recall*, these are calculated as follows:

$$Precision = \frac{TP}{TP + FP} \qquad Recall = \frac{TP}{TP + FN} \tag{5}$$

therefore

$$F1 = 2 \cdot \frac{Precision \cdot Recall}{Precision + Recall} = \frac{TP}{TP + \frac{1}{2}(FP + FN)} \tag{6}$$

In multi-class classification settings, an *F-score* is computed for each class (i.e. considering a binary classification problem of in-class vs. not in-class). The total *F-score*, named the *macro F-score* is then the average of the per-class F-scores. To cope with uneven class sizes, the *weighted F-score* is also computed, as a weighted average of the per-class F-scores, with the weights being proportional to the class sizes. Finally, the *micro F-score* is defined as the overall precision across all classes. The weighted macro F-score and micro F-score take into account the size of each class and are thus more appropriate for imbalanced classification problems.

**Normalised distance-based performance measure.**   For the evaluation of the rankings produced by the Persuasion Principle Optimal Ranking system we used the Normalised Distance-based Performance Measure (NDPM) presented in [53]. This metric evaluates the distance between two rankings. The performance of the proposed ranking is calculated by comparing the position of each item in both of them. This metric follows a rule set to determine if the items in the rankings are *agree*, *disagree* or are *compatible*:

- The orders $\succ_1$ and $\succ_2$ *agree* on items $i_1$ and $i_2$ if $i_1 \succ_1 i_2$ and $i_1 \succ_2 i_2$.

- The orders $\succ_1$ and $\succ_2$ *disagree* on items $i_1$ and $i_2$ if $i_1 \succ_1 i_2$ and $i_2 \succ_2 i_1$.

- The orders $\succ_1$ and $\succ_2$ are *compatible* on items $i_1$ and $i_2$ if $i_1 \succ_1 i_2$ and neither $i_1 \succ_2 i_2$ or $i_2 \succ_2 i_1$. That is when the items are tied or incomparable because one of them is not in the other order.

Following these rules, if the orders agree, the distance between them will not increase. Instead, if the orders disagree or are compatible, the distance will increase. Thus, as this metric calculates distances, the lower the score, the better the performance. The performance of our proposed methods has been evaluated using the evaluation measures of this section. Each measure validates a different aspect of the problem.

## Experimental scenarios

To evaluate and validate both individual and fused models, two specific scenarios have been employed, based on the final envisioned outcome to be predicted.

**First scenario.** The first scenario aims to analyse and predict each persuasion principle to identify exactly which strategy would be the most appropriate for each user. Starting with the application of each model separately and then their fusion, two evaluation sub-cases were performed: a) training and testing on the total of both datasets (cross-validation of GreenSoul and Prolific together with N = 743). b) Training only with GreenSoul data (N = 368) and testing only on the Prolific ones (N = 375). However, after testing both, the former outperformed the latter, hence results only for the best approach (a) are provided. The performance drop in the second approach might be because the two datasets (GreenSoul and Prolific) represent two different population distributions (even though we did the same type of screening on the Prolific platform as described in the section dataset). Our conclusion of this comparative experiment was the need to regularly retrain the models with more data to cope with differences in the user characteristics—overall if they wanted to be transferred to different contexts or latitudes. Or even if the people change their mindsets in the short-mid term. What is good about our final user features set, is that out of the nine questions, only three might change throughout the time: "Intentions, Confidence in technology, and PST".

**Second scenario.** The second main scenario aims to tackle the problem in a more high-level approach. The ten persuasion principles can be distinguished into two groups based on the type of interaction: persuasion principles that are driven by interaction with a system and the ones driven by interaction with people. Similarly, with the first scenario, the same two sub-cases are explored.

## Persuasion principle optimal ranking

For this approach, and considering that there have not been any state-of-the-art findings trying to achieve similar objectives, a benchmark with three different baselines to evaluate the performance of the persuasive strategy recommendation system has been designed. The first baseline is a random classifier that selects five random principles from the possible ten for each user. The second baseline, called 'Top5', proposed the average top 5 principles to all the dataset users. Finally, the last baseline is the proposed model itself, with the user features extracted from it. We employed a 5-fold cross-validation methodology to evaluate the baseline's performance and the model through all the data. In each fold, 80% of the data was used as the training set and the remaining 20% as the testing set. The average NDPM, as well as the standard deviation for each model for the testing set, are reported in Table 3.

The results reported in the table show that the model with the data performs better than the other proposed baselines. However, the Top 5 baseline also achieved good results in this experiment. As we analysed the dataset, it can be observed that there are five popular principles that most of the users liked the most. Even if these principles are not consistently ranked in the same position, the NDPM for a ranking that includes the same principles as the ground truth will always be better than a ranking that does not include one of those principles. Therefore, baseline results will always be high for this dataset because of the popularity of some principles and the similar rankings between many users. These findings are in line with the observation

**Table 3. NDPM and standard deviation (σ) for optimal ranking prediction model with the cross validation approach.**

| Baseline/Model Name | NDPM (σ) |
|---|---|
| Random | 0.50068 (±0,001) |
| Top 5 | 0.16358 (±0,003) |
| HiR without user data | 0.16646 (±0,002) |
| HiR | 0.15667 (±0,001) |

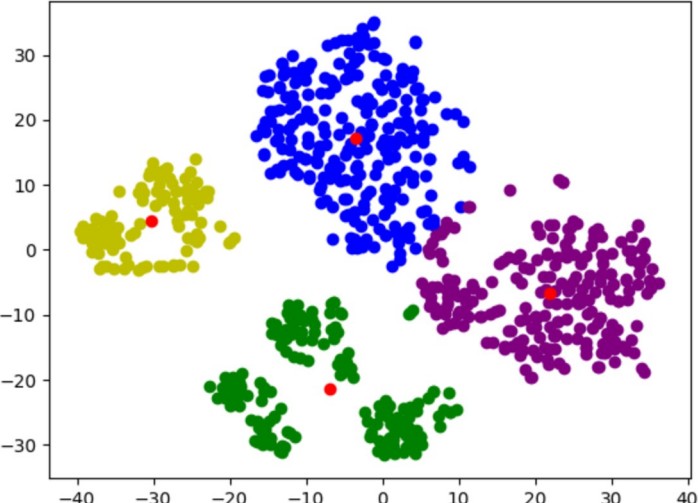

**Fig 4. T-SNE and K-means conducted to users' embeddings.** Red dots show the centroids of each of the clusters.

that four different user groups can be found who ranked the top 5 principles differently in the dataset that we used. Conducting a T-SNE to the user embeddings obtained from the model, these user groups can be easily clustered using agglomerative or simultaneous approaches. The latter was employed using K-means, from which we obtained four clusters that represent user profiles, as can be observed in Fig 4. Please, bear in mind that in the next Figure, unlike some other dimensionality reduction techniques (like PCA, where axes can represent principal components), the axes in a t-SNE plot do not have an inherent, interpretable meaning. Thus, they do not correspond to specific original features of the data. In essence, the focus of Fig 4 should be on the relative positioning and clustering of points, rather than trying to interpret the axes themselves. The axes serve as a means to visualize the high-dimensional relationships in a more comprehensible two-dimensional space.

## Persuasion principle score prediction

To evaluate the persuasion principle score prediction method, we have trained and tested the Random Forest regression (RF) and the Multi-Variate Regression (MVR) models, one for each principle, in 5-fold cross-validation, using the same folds as the ones used in the evaluation of the optimal ranking procedure. Furthermore, the usage of the same folds allows the comparison between the two methods and the fusion of their results. For the MVR model, the rank of the coefficient matrix has been set to 5, which achieved the highest accuracy scores.

The results of the evaluation are presented in Fig 5. The horizontal axis contains the target principles, while the vertical axis reports the RMSE errors of predicting the scores for each principle. The lines correspond to the mean RMSE across all cross-validation folds, for the RF and MVR methods, while the error bars correspond to the 95% inter-quantile range. We can see that the RMSE errors range from 1 to 1.5, indicating that the predictions deviate, on average, by 1–1.5 units from the true scores, which fall within the range of 1 to 5. We can also see that the RMSE values vary for all principles, with "Similarity" and "Social proof" achieving the lowest errors, suggesting more accurate predictions for these principles. The errors of the RF method are slightly lower than those of the MVR method, indicating that the RF method is more suitable for predicting individual principles, as expected given that MVR considers all

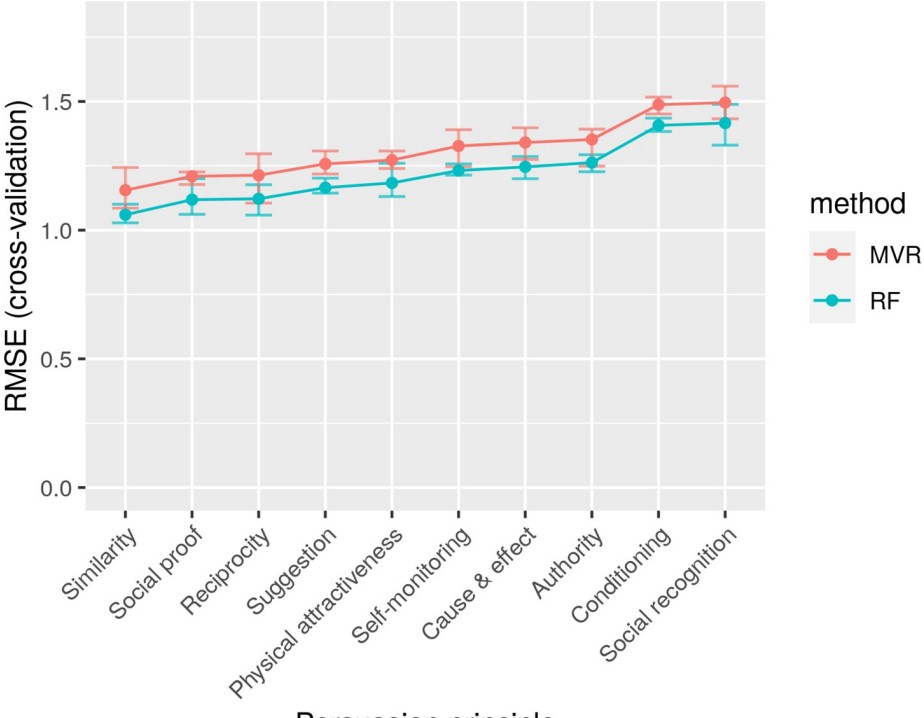

**Fig 5. Evaluation results for the proposed recommendation model, using the RF and the MVR models.** RMSE values are shown for all persuasion principles. The principles are sorted by increasing RMSE values.

principles simultaneously. However, it is important to note that even though MVR used a rank 5 coefficient matrix, it still achieved relatively high accuracy, especially when focusing on fewer "core" targets.

The importance of each predictor in the final predictions is shown in Fig 6. It can be seen that aspects of energy-related behaviour such as profile information, education and barriers, are among the most important predictors.

## Fusion of optimal ranking and score prediction

To evaluate the fusion of the two methods, we have first considered the problem of predicting the optimal persuasion principle to use for each user, i.e. the top-1 recommendation. Using the top-1 recommendation extends our previous work (where top-10 rankings and rating scores were computed) to bring it closer to the envisioned application of our methods, i.e. selecting the most appropriate strategy to use to persuade people. In a real-world setting, the system would need to select one strategy as the most appropriate, so being able to predict the top-ranked strategy is important. Considering only the top-1 principle makes the problem a multi-class classification problem so that measures such as F-score can be applied for evaluation. For example, if the ground truth for a user (i.e. the most appropriate principle) is "suggestion", we want to evaluate if the proposed methods will be able to rank "suggestion" as the top-ranked principle out of all considered principles. For the experimental scenarios, we used the fusion of the two methods to compute the top-5 principles and then used the top-1 principle as the prediction for the most appropriate persuasion principle for a user. Next, we compared this prediction to the ground truth, i.e. the top-1 principle reported by the users in the

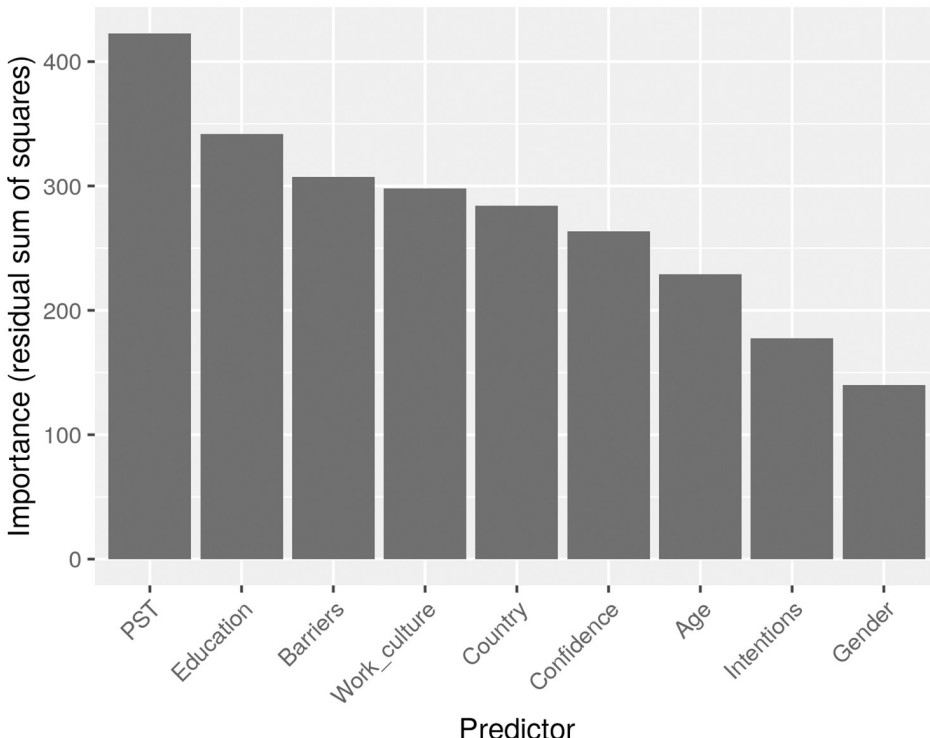

**Fig 6. Average importance of the predictors in the Random Forest regression models used for score prediction.**

questionnaires. As a baseline method for comparison, we have also used the most frequent top-1 answer in the training set to predict all users. In the majority of the cases, the top-1 strategy is: "Conditioning". Recall that the top-5 principles among the whole dataset were: "Conditioning", "Social-recognition", "Physical attractiveness", and "Self-monitoring".

The results are summarized in Table 4. The columns of the table correspond to the compared methods: most frequent (baseline prediction), optimal ranking individually, score prediction individually, and sequential fusion. For the score prediction, both the RF and the MVR models are used, so there are two pairs of columns for score prediction and fusion. The rows of the table correspond to the evaluation measures used: macro F-score, weighted macro F-score, and micro F-score.

We can see that for this type of problem, i.e. predicting only the top-1 result, the two individual methods perform rather poorly compared to the baseline of using just the most frequent result of the training set. This is more or less expected since the methods were trained with different goals in mind, i.e. predicting the complete ranking and the whole set of scores, respectively. However, when the two methods are fused sequentially, they surpass the baseline,

**Table 4. Comparison between the individual methods and the fused methods, for predicting the top-1 persuasion principle.**

| Evaluation measure | Most frequent top-1 | Optimal ranking | Score prediction (MVR) | Fusion (MVR) | Score prediction (RF) | Fusion (RF) |
|---|---|---|---|---|---|---|
| Macro F-score | 0.039 | 0.023 | 0.025 | 0.114 | 0.033 | **0.118** |
| Weighted macro F-score | 0.097 | 0.045 | 0.016 | 0.236 | 0.028 | **0.246** |
| Micro F-score | 0.244 | 0.115 | 0.030 | **0.316** | 0.041 | 0.315 |

providing better predictions, as verified by all the reported measures. The fact that the fused method (ensemble) performs better than the individual methods is a recurring pattern in the ensemble learning literature [54]. Combining different predictors increases the confidence in the predictions and reduces the error. In our case, keeping the top-5 principles from the first method probably acts as a filter that removes "noisy" principles that would reduce the accuracy of the second method. In this manner, the two methods act complementary, achieving better results when combined rather than when used individually.

Comparing the Random Forest (RF) and multi-variate regression (MVR) models for score prediction, we can see that the RF model achieves higher performance than MVR, suggesting that it can more accurately predict the top-1 principle. However, the fused models achieve comparable performance, which shows that fusion is able to overcome the shortages of even less accurate models.

Overall, although the numbers themselves are low, the results show that the user information encoded in the predictor variables contains information that can be used to provide better and personalised recommendations rather than recommending the same strategy for all users.

As a second experimental scenario, we have also evaluated the performance of the models in predicting the top-1 type of interaction, according to the types of interaction reported in Table 1. After we computed the top-1 principle, as in the evaluation above, we used Table 1 to map each principle to the corresponding interaction type and re-computed the evaluation measures for this type of outcome. There are only two outcomes this time, interaction with people and interaction with a system, i.e. the goal is to predict which kind of interaction would be most appropriate for each user.

The results are presented in Table 5. The absolute numbers are now larger than those for predicting the exact principle, which is expected since the number of classes is now only two. We can also see that again, the fused method performs better than either of the two proposed methods when used individually and also better than the baseline method where the most frequent type of interaction is used as the same prediction for all users. The RF model is again more accurate than the MVR model, although to a lower degree, while the fused models are again comparable in performance.

## Discussion

There is a recurrent interest in studying the persuasive potential of personalization. In this article, two AI-based approaches are proposed and then fused to this tailoring purpose. Whereas each approach can work in isolation, and their assessment can be provided separately, the authors of this paper firmly believe that the ensemble proposed is a better alternative to assess each persuasion strategy's impact on each specific individual. Regarding the main results obtained from the model, we have observed that the main predictors (features) for the self-reported (preferred) persuasion strategy according to the dataset are PST (attitudinal profile related to the way energy is used in the workplace), the level of Education, the Barriers encountered to behave energy-efficiently and the Work culture. This outcome was tested by using

**Table 5. Comparison between the individual methods and the fused methods, for predicting the top-1 type of interaction.**

| Evaluation measure | Most frequent top-1 | Optimal ranking | Score prediction (MVR) | Fusion (MVR) | Score prediction (RF) | Fusion (RF) |
|---|---|---|---|---|---|---|
| Macro F-score | 0.374 | 0.338 | 0.449 | 0.592 | 0.455 | **0.592** |
| Weighted macro F-score | 0.450 | 0.344 | 0.460 | 0.610 | 0.465 | **0.611** |
| Micro F-score | 0.599 | 0.481 | 0.461 | **0.616** | 0.462 | 0.613 |

random forest and extra trees for the calculation of the importance per feature. Less important predictors are found to be the age, the gender or the intentions to behave pro-environmentally. This means that to be effective, the model proposed in this article needs input user data which is closer to the context in which the cues will be elicited (in our case, the workplace) and less information that can be more general. Regarding the persuasive principles, two main results are derived from this research. The first one is that the top-rated persuasive principles using a descriptive analysis were "Conditioning", "Social-recognition", "Physical attractiveness", "Self-monitoring", and "Reciprocity". However, when it comes to finding the principles that suit best each user, the ones with lower prediction errors according to Fig 5 were "Similarity", "Social-proof" and "Reciprocity", "Suggestion" and "Physical attractiveness". Therefore, we can observe how the model presented in this manuscript aims at personalization instead of generalization. In essence, providing the user with the top-rated principles derived from the accumulation of all answers collected is not the way to go, but to understand better the user's traits to successfully suit the persuasive principle for them nonetheless. Regarding the type of interaction of the persuasive principle (People vs. System from Table 1), doing an average, both the statistical descriptive analysis and the results of our model revealed that both types are almost equally preferred. This balance implies a need for persuasive systems to integrate both human-centric (social proof, recognition) and system-centric (self-monitoring, suggestions) elements. In the following, we discuss the main results separated by sub-sections.

## Enhancing the recommendation through a cascade of models

In the ensemble model provided in this paper, the former computational recommender system serves as filtering for the latter. That is, the first model can provide the order of preference that each user will have on the ten persuasive principles upon evaluation (see Table 1). Indeed, according to the Normalized Distance-based Performance Measure (NDPM), the first model outperforms the random assignment of persuasive principles and even the best top-5 principles to newcomers. However, having a ranking is insufficient to understand if the top principles will have a tangible impact on the user. Therefore, the latter model uses the top 5 five principles learned from the first model instead of all the ten initially evaluated (i.e. the first model can be understood as a dimensionality reduction approach). With these measures, the optimal scoring for each persuasion strategy is more likely to be obtained. Still, researchers using this model can fine-tune it to find better performance with, for example, a selection of a smaller subset of persuasion principles and facilitate the final selection of the latter model. The proposed model achieves an RMSE around 1–1.5 on a scale from 1 to 5, which, although not too low (relatively to the range of the score it is about "average"—not too poor or too good), is low enough to make the model useful in a real setting. In practical terms, this RMSE range suggests the model can provide useful, albeit not perfect, insights or predictions within the given context. While our model outperforms some benchmark models, the main goal of this paper was not purely to obtain the best performance metric, but mainly to propose a novel way to address the way messages are presented to end users to ensure that the desired action (e.g., switching off the PC while doing breaks) will eventually occur. With this ensemble, we demonstrate that cascading models seem to be a way to effectively address end-user tailoring. It is also worthwhile to notice that the model accuracy is different for different principles, with principles such as "similarity" and "social" proof achieving more accurate predictions. This means that when the model predicts these principles as most appropriate, it can be more confident about the correctness of this decision. Such an indication of confidence can also be of value to the policymaker when deciding how to interpret the model outcomes.

## Framing the dimensions of persuasion per individual

To optimise the model in terms of the type of persuasion (see Table 1) that can be more effective to a specific user, an outcome is provided in terms of the type of motivational messages each user prefers between a more social-based interaction and system-based interaction. This idea is in line with existing literature where some authors explore the use of social practices for persuasion [55] and others use more system-oriented approaches such as visual interaction [56]. Having this outcome for each individual, scholars, researchers or intervention designers can have a clearer idea of what kind of contextual feedback to provide to end users in order to become more energy efficient (e.g., lowering their energy footprint) in tertiary buildings. This is an important outcome as people do not react equally to the same stimuli. Therefore, our method can provide hints of the types of interaction of messages that can be more relevant to a user depending on their personal traits. Some would prefer a message such as "your peers are already lowering their energy footprint" (i.e., social proof- People) while others would prefer a message such as "the system allows you to check your energy performance at any time" (i.e., self-monitoring—System). In essence, if instead of looking for the best persuasion strategy, a recommender system would like to know what kind of context is the most valuable to present a message to end-users, the model we presented in the paper can help to provide that with higher reliability.

## Applying the model in real-world scenarios

The majority of the prior work reviewed conducted qualitative (focus groups, interviews) or quantitative research (questionnaires) to elucidate if certain socio-economic characteristics affect the persuasion potential of persuasive strategies [57, 58]. This is mostly done through the use of observational studies, structural equation models and systematic reviews of the literature [59–61]. However, when it comes to providing different persuasive strategies depending on the context, or the user traits, or depending on the success obtained from the previous experiences nudging a behaviour, the authors of this research have realised that the literature is scarce. We therefore aim to shed light on this area with the approach presented. Our main statement has as its basis the fact that the user archetypes are dynamic and changing. Consequently, if someone can obtain explicit information from users at certain moments, it will be much more effective to tailor a message to them [62]. The fact that the amount of information needed by the proposed method is not vast (only 9 features) makes it easier to get repeated measures throughout the time to capture such a dynamicity (e.g., using an app that facilitates the collection of data through rapid micro-surveys or online questionnaires without putting much burden to the user). Moreover, the fact that the context of the suggestions is personalised would guarantee that the suggestions keep up with the evolving user profiles. Therefore, we argue that the use of intelligence-based approaches for this purpose is novel. The only similar approaches we learned from the body of research are Kaptein's persuasive profiles-based [59], Mogles' agent-based models based on internal values [63] or Kang's virtual healthcare [64]. However, computationally speaking, our approach seems to be more easily adaptable to real online services and mobile web or social apps. In essence, the proposed method could be used in a real setting as the back-end engine of a persuasion recommendation system, that would take this user information as input and compute the most appropriate strategy for each user. In this regard, other research works underscore the effectiveness of personalized normative messages in recommender systems for promoting energy conservation [65]. This aligns with the emphasis on contextualized persuasion principles, substantiating our approach with empirical evidence. Moreover highlights that tailored messages based on social norms can significantly influence user behaviour in energy efficiency. For this reason, applying this strategy

would then require implementing each of the persuasion principles using specific actions, such as motivational messages (e.g., similarity or suggestion principles) and/or monitoring applications (e.g., self-monitoring or cause-effect principles), through web or mobile interfaces. This will be the topic of future endeavours and experimentation and we expect that other scholars can use our work to test these or other ideas in their research overall when aiming at understanding which persuasive principle elicits behaviour change in the mid and long-term.

In essence, it is important to address the practical challenges and considerations that may arise during the implementation of our proposed framework in real-world settings. Firstly, integrating our system into existing infrastructures requires careful planning and adaptation to ensure compatibility with different technological environments across the workspace. Secondly, collecting and handling user data, which is a cornerstone of our framework, must navigate the complex landscape of privacy laws and ethical considerations, as we will explore in the next sub-section. Thirdly, the effectiveness of our system depends on user engagement and acceptance. Additionally, the scalability of the system must be considered to ensure that it remains efficient and effective as it adapts to varying numbers of users and different environments. Finally, ongoing monitoring and evaluation are necessary to identify and address any unforeseen issues after deployment, and to ensure the system's relevance and effectiveness in dynamically changing real-world scenarios as the workplace.

## Privacy and Ethical considerations and model generalization

From a practical point of view, the fact that the proposed method can predict optimal personalised persuasion principles from a few input features makes it suitable for use in real-world applications. In essence, by just obtaining nine features of the user (e.g., gender, education or willingness to do green actions), our approach is able to provide bespoke messages to the users to motivate them to continue doing green practices related to energy efficiency in tertiary buildings (e.g. switching off a lamp when not in use, use the low energy mode in their PCs, or nudge people to change equipment to more energy efficiency options). However, special attention should be given to the privacy, legal and ethical aspects if the proposed system would be deployed in real-case scenarios. Another issue somehow connected to the previous, is that some people can be reluctant to provide personal data in a micro-survey (e.g., their age (although we asked the age range, as can be seen in the surveys uploaded to Zenodo), the country where they live, their gender or the education level). Others can find it cumbersome to answer a questionnaire several times asking for their intentions to behave pro-environmentally or their energy consumer archetype—PST—(these two constructs from Table 2 have an instrument of 12 and 9 items behind). Therefore, from the nine user features, we consider that only Intentions, Confidence in technology and PST can change over time whereas the rest are more static. In this regard, we challenge the research community to find shorter, yet still valid, instruments that can ensure that these constructs are measured correctly. Another option to lower the burden on the user would be to create a new model based only on the top user traits (i.e., from Fig 6: PST, Education, Barriers and Work culture) to see if still the model is effective predicting the preferred persuasive strategy or to create a new model with only static features so no repeated measures are needed as time passes. Another option would be to collect part of these data indirectly and unobtrusively through monitoring devices in the workplace (e.g., barriers or work-culture). The main issue that appears, apart from the extra cost, is the privacy concerns that installing new equipment would pose to the employees in the workplace [66, 67]. Again, the future of this work is in the direction of finding ways to create data-driven user profiles [68] without burdening the end-user and also preserving their privacy and ethical

concerns at all levels (e.g., by using federated learning to separate the datasets per country as we did in a previous work [26]. It is worth mentioning that the Country where people live may affect their acceptance of energy-efficiency interventions [69]) and it should be taken into account to measure the effectiveness of a potential intervention based on the proposed system. Finally, in terms of generalization, our model indeed leans on data for a specific field of application (i.e., energy efficiency) and a specific context (i.e., the workplace). Nevertheless, we argue that the model is valid for any intervention related to sustainability issues (e.g., saving water or other resources, or improving the air quality). The only user feature that has to be removed as a predictor is "Work-culture" which is among the most relevant features (see Table 2) to accurately predict the persuasive principle. An option to tackle this issue would be to find another context-based predictor that will retain similar importance to ensure adequate inference.

## Nudging vs. persuasion

While persuasion in literature employs strategic communication to influence beliefs and actions, nudging subtly steers behaviour in a desired direction without necessarily applying direct persuasion. As discussed by Wecker, Kuflik, and Stock [70], while not mandatory, both persuasion and digital nudging benefit from personalization, opening new opportunities for enhancing user experiences and bespoke interaction when used appropriately. According to Herrick [71], persuasion, in its broader sense, is an art of rhetoric, using specific formations to achieve desired effects on audiences. This can involve both rational and emotional appeals, influencing individuals' knowledge, beliefs, and interests. Nudging, as defined by Thaler and Sunstein [72] and by Mols et al. [73] is identified as a distinct mode of governance, aiming to modify behaviour subtly without explicit persuasion strategies or incentives such as pecuniary ones. It uses choice architecture to make certain behaviours more likely without restricting freedom of choice. In this manuscript, we are focused on persuasion principles to influence energy-behavior activities. However, our approach indeed has a lot of similarities with nudging, as we are also trying to enact certain behaviours in a bespoke manner, aligning with insights from Eslambolchilar and Rogers [74] on how technology can leverage persuasion and nudging".

## Limitations of the current work

One of the most common limitations of behavioural studies is the sample size. In a recent systematic review [6] on the energy domain specifically, all of the 38 articles that have been extracted from the literature ranged from 4 to 651 individuals, introducing quite the doubt for the results. Although the sample size of this study is larger (N = 743), the population they belong to is very wide and therefore, more evidence is needed to observe if the results can be generalised to other actors, societies or cultures. Recall that the data from this study is obtained from four countries: Austria, Spain, Greece and the UK. The diversity in energy behaviours and cultural differences beyond these four European countries limits the extent to which our findings can be applied globally. Therefore, caution must be exercised in generalizing our results to populations in diverse geographic and socio-cultural contexts outside of Europe. This limitation underscores the need for further research in varied settings to enhance the universality of our findings. More particularly, provides us with a line for future research to remove the location factor (country) and explore if our computational model works better when people belong to the same society/culture.

The second major limitation of this study is that the factors/features obtained for training the presented models are tightly coupled to the energy efficiency context in the workplace (e.g.

if the people would be willing to join an environmental campaign in the workplace, the barriers that people encounter in their workplace to act pro-environmentally, their current occupation or their behavioural archetype. Thus, our proposal is dependent on whether the end-users are considered as pinballs, shortcuts, or thoughtful which are stereotypes related to the way they use energy resources [43]. Therefore, to fit the model to other contexts beyond the workplace and energy efficiency, it is advisable to find the socio-economic and behavioural characteristics that are context-specific for the new field of application to be then linked to the selection of persuasive principles studied in this paper. One option, if the objective is to stick to the generalist approach (not case-specific), could be to try to link the persuasion principles to personal traits such as those obtained from validated questionnaires such as Big Five (Ocean) or DiSC [75].

Another limitation of this work is that it relies on the subjective appropriateness of the persuasion principles, as reported by the users themselves in the provided questionnaires. The end-users may have certain misconceptions about what would affect their behaviour, which are imposed by pre-existing beliefs or social influence that do not reflect reality [76]. Additionally, the subjective nature of the self-reported data in our study is a significant limitation, as it can introduce biases related to personal perceptions, potentially affecting the accuracy and reliability of our findings. This aspect is crucial in understanding the implications and applicability of our results. In the current work, subjective perceived appropriateness was used as a proxy for actual appropriateness, due to the current lack of an extensive study to evaluate the actual appropriateness. Such a study would require applying multiple persuasion principles to the same user over a large time span and monitoring their effects in terms of energy savings, through extensive monitoring. This is a plan for future endeavours. However, the results of the current study are a first step towards this direction.

Regarding the persuasion principles studied in this paper, the authors acknowledge that their selection of ten principles was based on expert knowledge and from a survey conducted during the process of a research project to observe which were most prevalent among respondents. However, putting together the principles from Cialdini (6 principles), Fogg (15) and Oinas-Kukkonen (25) aforementioned in this paper and adding Abraham & Michie (93) [77], a researcher will find him/herself with more than one hundred of options to evaluate. A better approach than the one followed in this article would have been to obtain the most effective principles in the context of sustainability reporting in the body of knowledge. Still, the evidence of success or failure is not always provided in the same way, making the meta-analyses on this field challenging.

Another limitation of the presented recommender system presented against other computational models reviewed is that our approach is not framed nor linked to any behavioural framework (e.g. Transtheoretical Model [78], Belief-Values-Actions [79], Elaboration Likelihood Model [80], Social cognitive theory [81] or Theory of Planned Behaviour [82]). Therefore, future work should be focused on expanding the cause-effect idea that the model proposes towards providing specific persuasive messages to determinants of behaviours based on existing behavioural frameworks.

Finally, whereas the results are enhanced in the two-step pipeline proposed with regard to the simplest isolated models, the outcomes reported are not as conclusive as the authors would have expected when they designed the study. Nevertheless, they provide room for improvement by exploring other features that might be more relevant when inferring the most impactful persuasive strategy for a specific user based on previous knowledge and context. For instance, the model ensemble approach presented in this paper opens the possibility of studying alternative options to combine the outputs of both models. In this way, together with the ranking obtained in the Persuasion Principle Optimal Ranking, additional scores could be

obtained and combined with the scores of the Persuasion Principle Score Prediction stage. For that, a multi-objective optimization strategy could be applied in order to search for optimal solutions that maximize the output of both models and balance their trade-offs. Pareto multi-objective optimization analysis [83] could be applied to find the small-sized subset of solutions with the best performance [84]. This provides an example of the flexibility of the proposed approach.

## Conclusion

This paper has presented the design and assessment of a recommendation framework that aims towards identifying the optimal persuasion strategy per user based on self-reported data towards delivering energy-related recommendations to end-users in buildings. It should be clarified that the framework does not report a one-size-fits-all strategy that can be used to persuade users. Instead, it is a system that predicts which is the optimal strategy for each user, which may be different for different users. In this sense, the evaluation of the method was not based on the effects of following a suggested strategy but rather on whether the suggested strategy for a user is the one that the user would pick as the most effective. We deem that in real-life situations the decision-making of an action is usually quick and binary: we either do it or we do not do it. Therefore, this paper has put the focus on persuasive principles that may work best for any single user. The presented engine performs using as input a simple survey of 9 constructs (predictors) and selects from ten well-known persuasion principles that would be the most effective to the examined user. The system has been evaluated and validated through an online survey performed in four countries across Europe, assessing about 743 users. It delivers promising insights such as finding a reduced set of user features (i.e., user behaviour type, and Education) that predict the preferred persuasive principles for users. Furthermore, the proposed approach can infer the top persuasive principles for a user based on user self-reported data. The results show that our model performed properly with some principles such as "Similarity", "Social Proof", or "Reciprocity". Our outcomes are supported by increased accuracy results compared to non-personalised baseline methods (e.g., the top-5 most-rated persuasive principles among respondents). Furthermore, the engine manages to recommend to each user either a single most appropriate persuasive principle or a more generic type of interaction: System vs People. The former can employ existing ICT-based systems that require more cognition and interaction with the system. the latter is conceived for interaction with other people, colleagues or peers in a social manner. By delivering the most effective persuasive strategy in a personalised manner, it is expected that long-term results can be achieved in terms of energy awareness and occupant behaviour in working environments. It is also expected that such an engine can complement current recommender systems, delivering even better results in terms of energy efficiency, awareness, engagement, and overall user experience.

## Supporting information

**S1 File. Survey instrument.** Validation Profile Questionnaire.
(PDF)

## Acknowledgments

We would like to thank the reviewers for helping us to ameliorate the manuscript.

## Author Contributions

**Conceptualization:** Rubén Sánchez-Corcuera, Diego Casado-Mansilla, Apostolos C. Tsolakis.

**Formal analysis:** Ilias Kalamaras, Apostolos C. Tsolakis.

**Funding acquisition:** Stelios Krinidis, Cruz E. Borges, Dimitrios Tzovaras, Diego López-de-Ipiña.

**Investigation:** Rubén Sánchez-Corcuera, Apostolos C. Tsolakis, Cruz E. Borges.

**Methodology:** Ilias Kalamaras, Diego Casado-Mansilla, Apostolos C. Tsolakis, Oihane Gómez-Carmona.

**Project administration:** Dimitrios Tzovaras, Diego López-de-Ipiña.

**Software:** Ilias Kalamaras.

**Supervision:** Diego Casado-Mansilla, Apostolos C. Tsolakis.

**Validation:** Diego Casado-Mansilla.

**Visualization:** Ilias Kalamaras, Rubén Sánchez-Corcuera.

**Writing – original draft:** Ilias Kalamaras, Diego Casado-Mansilla, Apostolos C. Tsolakis.

**Writing – review & editing:** Diego Casado-Mansilla, Apostolos C. Tsolakis, Oihane Gómez-Carmona.

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
