## [Decision Letter · Decision Letter 0]

30 Oct 2023

PONE-D-23-30762A cascading model for nudging employees towards energy-efficient behaviour in tertiary buildingsPLOS ONE

Dear Dr. Casado-Mansilla,

Thank you for submitting your manuscript to PLOS ONE. After careful consideration, we feel that it has merit but does not fully meet PLOS ONE’s publication criteria as it currently stands. Therefore, we invite you to submit a revised version of the manuscript that addresses the points raised during the review process. The crucial seems overcoming Reviewer #2 objections.

We look forward to receiving your revised manuscript.

Kind regards,

Krzysztof Malarz, D.Sc., Ph.D., M.Sc.

Academic Editor

PLOS ONE

"Ministry of Economy, Industry and

Competitiveness of Spain for IoP, under Grant No.: PID2020-119682RB-I00"

"This work is partially funded by the European Union’s Horizon 2020 Research and

Innovation Programme through the projects WHY (Grant Agreement No. 891943),

SOCIO-BEE (Grant No.: 101037648), and GREENGAGE (Grant No.: 101086530).

Furthermore, we express our gratitude to the German Federal Ministry of Education

and Research (BMBF) and the Greek General Secretariat for Research and Technology

(GSRT) in the context of the Greek-German Call for Proposals that funded the

SIT4Energy project. Besides, we acknowledge the Ministry of Economy, Industry and

Competitiveness of Spain for IoP, under Grant No.: PID2020-119682RB-I00"

"Ministry of Economy, Industry and

Competitiveness of Spain for IoP, under Grant No.: PID2020-119682RB-I00"

7. In your Data Availability statement, you have not specified where the minimal data set underlying the results described in your manuscript can be found. PLOS defines a study's minimal data set as the underlying data used to reach the conclusions drawn in the manuscript and any additional data required to replicate the reported study findings in their entirety. All PLOS journals require that the minimal data set be made fully available. For more information about our data policy, please see http://journals.plos.org/plosone/s/data-availability.

8. We note that you have stated that you will provide repository information for your data at acceptance. Should your manuscript be accepted for publication, we will hold it until you provide the relevant accession numbers or DOIs necessary to access your data. If you wish to make changes to your Data Availability statement, please describe these changes in your cover letter and we will update your Data Availability statement to reflect the information you provide.

Reviewers' comments:

Reviewer's Responses to Questions

**Comments to the Author**

1. Is the manuscript technically sound, and do the data support the conclusions?

Reviewer #1: Yes

Reviewer #2: Partly

Reviewer #3: Yes

2. Has the statistical analysis been performed appropriately and rigorously? 

Reviewer #1: Yes

Reviewer #2: No

Reviewer #3: Yes

3. Have the authors made all data underlying the findings in their manuscript fully available?

Reviewer #1: Yes

Reviewer #2: No

Reviewer #3: Yes

4. Is the manuscript presented in an intelligible fashion and written in standard English?

Reviewer #1: Yes

Reviewer #2: Yes

Reviewer #3: Yes

5. Review Comments to the Author

Reviewer #1: This research designs a recommendation framework to identify individualized persuasion strategies that encourage energy-efficient behaviors in building occupants. Using a large dataset from four European countries, the framework ranks ten expert-preselected strategies via matrix factorization and random forest regression models. It demonstrates higher accuracy than non-personalized methods and offers insights into the most effective strategies based on user data.

Overall, the paper needs signficant improvements on the following aspects:

1) The literature review of this article is very terse. Many recent studies discussing energy consumption behavioral change has not been discussed in the literature review. The authors should elaborate more on this aspect by discussing the following studies: Smart Sensing and End-User Behavioral Change in Residential Buildings: An Edge Internet of Energy Perspective; Is there any room for renewable energy innovation in developing and transition economies? Data envelopment analysis of energy behaviour and resilience data; The emergence of explainability of intelligent systems: Delivering explainable and personalized recommendations for energy efficiency; Intelligent edge-based recommender system for internet of energy applications; Understanding Waste Management Behavior Among University Students in China: Environmental Knowledge, Personal Norms, and the Theory of Planned Behavior; A novel approach for detecting anomalous energy consumption based on micro-moments and deep neural networks; Reshaping consumption habits by exploiting energy-related micro-moment recommendations: A case study; REHAB-C: Recommendations for Energy HABits Change; A micro-moment system for domestic energy efficiency analysis; Using big data and federated learning for generating energy efficiency recommendations; Smart fusion of sensor data and human feedback for personalized energy-saving recommendations; IoT based smart and intelligent smart city energy optimization

2) Kindly fix the formatting problem in page 2.

3) While the sample size of 678 users across four countries is commendable, the authors should discuss the extent to which the findings are generalizable to wider populations, especially outside Europe.

4) The self-reported nature of the data can introduce biases. The authors might consider addressing this aspect and discussing potential limitations of the self-reported data.

5) The paper briefly mentions increased accuracy over non-personalized methods but lacks an in-depth comparison or analysis. This comparison would be beneficial for readers to gauge the actual improvement the proposed framework offers.

6) The conclusion highlights the potential of the framework, but there is a lack of discussion about the practical challenges or considerations in deploying such a system in real-world settings.

Reviewer #2: The authors propose a cascading/fusion approach to better predict the best nudging/persuasion strategy for individual users in the context of energy efficiency. The presented analysis is based on survey data, which was partly previously collected and partly collected in the context of the present research. The reported experiments indicate that combining the outcomes of two prediction models is synergistic and leads to rankings that are closer to the "ground-truth" ranking than other approaches (a random one and a popularity/frequency-based one).

The work is generally well prepared and easy to follow. The topic itself is relevant. The discussion of related works seems appropriate.

Overall, however, I think the contribution to the existing body of literature of this work seems too limited. The work extends the authors' own previous works, and the main addition seems to lie in the combination of two prediction models (and some additional data that is collected). The outcome of the combination is positive. For a journal publication, I would expect a much deeper analysis. I appreciate that the authors discuss limitations of their work in various respects. Still, I would have hoped that the paper delivers more than trying out some combination of predictors on a dataset (which also has some major limitations).

Unfortunately, the authors decided to not share the code and the data, which makes it impossible for others to validate the findings. The authors argue that they do not want to do this for privacy/ethical reasons, but it remains unclear why it is impossible to a) share the code for the analyses and b) share anonymized data.

Other remarks:

1) There is some difference between nudging and persuasion, according to the literature. Should be discussed.

2) It would be helpful if the distribution of top-1 ranked strategies in the original data would be shown. In general, a deeper analysis of the underlying data seems advisable.

3) The data itself, as mentioned by the authors, might be rather noisy or even biased, depending, e.g., how things were interpreted by the participants.

4) Pre-processing and feature selection: The authors perform feature selection for a given classification task, but the classification task is not explicitly described.

5) The threshold of 80% of keeping data seems a bit arbitrary.

6) "which are then fused sequentially to get aggregated results" -> it is not so clear at this stage what "fused sequentially" would mean.

7) "Specifically, it extends MF by including the latent factors of users (socio-economic features) and items" -> The paper is not self-contained here. It is also not clear at this stage what the elements of the matrix are which should be factorized.

8) Fusion: A variety of list combination strategies are possible. Why was this particular fusion strategy applied?

9) Experiments: The assumption of several analyses is that only one top-ranked strategy per user. In reality, also considering noise, there might be two or more equally suited strategies per use. The evaluation approach thus seems a bit narrow. Could be discussed.

10) Experimental scenarios: Two datasets are used, leading to different results. A deeper investigation is missing. Also, I wonder if it is meaningful to compare the used metrics on an absolute scale.

11) Figure 3 and the corresponding analysis: Why is the NDPM used here as the only measure? Why are different measures used in the other experiment? Why not reporting F1 here as well?

12) "Conducting a T-SNE to the user embeddings obtained from the model" -> Where do the user embeddings come from here?

13) "meaning that the predictions are an average of 1-1.5 units away from the true scores" -> Is this true? It is not the MAE, which can be easily interpreted in absolute terms, but here the authors use the RMSE.

14) Table 4: Now here we use different forms of the F-score. It is quite surprising that the combination of two rather weak predictors (eg 0.02 and 0.03) leads to a combined value that is multiple times higher (0.11). Unfortunately, no code and data is provided to verify this rather surprising result.

Minor:

The writing is a bit wordy at times, starting with a quite long abstract.Jesse et al. -> Jesse and Jannach"is low enough to make the model useful in a real setting." -> How do we know?Starke et al. worked on nudging in the energy efficiency domain, e.g. [a]. Their works could be discussed.

[a] Alain D. Starke, Martijn C. Willemsen, Chris Snijders: With a little help from my peers: depicting social norms in a recommender interface to promote energy conservation. IUI 2020: 568-578

Reviewer #3: The manuscript is well written and is structured well. The research aims at providing data-driven personalized approaches for persuasion towards energy efficient behavior. The manuscript demonstrates robust statistical analysis and provides a good description of the methodology derived form the previous works.

A few suggestions to improve the paper-

1) Discuss what are the implications on different stakeholders- e.g. facility managers (optimized operations, lesser complaints?), occupants (requires more engagement or understanding of system/people), owners (operating costs), etc.

2) Were there any methods applied to test reporting biases?

3) How do you envisage this framework can be replicated in specialized settings - such as schools, senior living homes etc.

4) Can you comment on the micro f-score being slightly better for MVR than RF ( for fusion) in Tables 4 and 5?

5) Figure 3 What does the x and y axis represent?

6. PLOS authors have the option to publish the peer review history of their article (what does this mean?). If published, this will include your full peer review and any attached files.

Reviewer #1: No

Reviewer #2: No

Reviewer #3: **Yes: **Jeetika Malik

---

## [Author Response · Author response to Decision Letter 0]

25 Jan 2024

Reviewer 1

Overall, the paper needs significant improvements on the following aspects:

1. The literature review of this article is very terse. Many recent studies discussing energy consumption behavioral change has not been discussed in the literature review. The authors should elaborate more on this aspect by discussing the following studies: Smart Sensing and End-User Behavioral Change in Residential Buildings: An Edge Internet of Energy Perspective; Is there any room for renewable energy innovation in developing and transition economies? Data envelopment analysis of energy behaviour and resilience data; The emergence of explainability of intelligent systems: Delivering explainable and personalized recommendations for energy efficiency; Intelligent edge-based recommender system for internet of energy applications; Understanding Waste Management Behavior Among University Students in China: Environmental Knowledge, Personal Norms, and the Theory of Planned Behavior; A novel approach for detecting anomalous energy consumption based on micro-moments and deep neural networks; Reshaping consumption habits by exploiting energy-related micro-moment recommendations: A case study; REHAB-C: Recommendations for Energy HABits Change; A micro-moment system for domestic energy efficiency analysis; Using big data and federated learning for generating energy efficiency recommendations; Smart fusion of sensor data and human feedback for personalized energy-saving recommendations; IoT based smart and intelligent smart city energy optimization. 

Thank you very much for your valuable suggestion. We acknowledge the necessity to expand our literature review to incorporate recent studies addressing energy consumption behaviour change. In response to your suggestions, we have revised the Introduction section to include key studies that align with our research scope.

Among others, we have integrated references like "Smart Sensing and End-User Behavioral Change in Residential Buildings: An Edge Internet of Energy Perspective" and "Is there any room for renewable energy innovation in developing and transition economies?" to emphasize the evolving landscape of energy consumption and efficiency. Additionally, we have discussed studies focusing on "The emergence of explainable intelligent systems" and "Intelligent edge-based recommender systems for the Internet of energy applications" to underline the advancements in personalized and intelligent energy-saving systems.

Moreover, we have included insights from "Understanding Waste Management Behavior Among University Students in China" and "A novel approach for detecting anomalous energy consumption" to showcase diverse aspects influencing energy behaviour. The inclusion of studies on micro-moments and federated learning further enriches our discussion on innovative approaches to energy efficiency.

These additions provide a more comprehensive view of the current state of research in energy-saving behaviours and recommendation systems. We believe that these enhancements significantly improve the literature that contextualizes our manuscript. 

2. Kindly fix the formatting problem in page 2..

Thank you for bringing this to our attention. The manuscript has been thoroughly reviewed to address potential issues.

3. While the sample size of 743 users across four countries is commendable, the authors should discuss the extent to which the findings are generalizable to wider populations, especially outside Europe.

We can not agree more with the reviewer's assessment. The concern regarding the self-reported nature of data introducing biases is valid and acknowledged. For this reason, in the "Limitations of the current work" section, the manuscript addresses various constraints, including sample size and population diversity, which may affect the generalizability of the results across different societies and cultures. The limitation concerning the context-specific factors/features for training the models and their dependency on energy efficiency in the workplace is also discussed. Additionally, the manuscript recognizes the subjective nature of user-reported appropriateness of persuasion principles and the challenges in evaluating their actual effectiveness. Future research directions are proposed to address these limitations, including expanding the study to different contexts and linking the persuasion strategies to well-established behavioral frameworks. 

Nevertheless, the "Limitations of the current work" section of the manuscript has been updated to acknowledge this concern more explicitly:

The diversity in energy behaviors and cultural differences beyond these four European countries limits the extent to which our findings can be applied globally. Therefore, caution must be exercised in generalizing our results to populations in diverse geographic and socio-cultural contexts outside of Europe. This limitation underscores the need for further research in varied settings to enhance the universality of our findings. More particularly, provides us with a line for future research to remove the location factor (country) and explore if our computational model works better when people belong to the same society/culture.

4. The self-reported nature of the data can introduce biases. The authors might consider addressing this aspect and discussing the potential limitations of the self-reported data.

Once again, we fully agree with the reviewer's suggestion. Self-reporting allows participants to provide information based on their personal experiences and perceptions, which is vital in behavioural studies like ours that focus on subjective aspects such as personal attitudes and beliefs. While self-reported data can be influenced by biases, it offers valuable insights into individual behaviours and motivations that are often difficult to measure through observational methods alone. 

In any case, we recognize the limitations of using self-reported data in our study and, in response to their recommendation, we have revised the "Limitations of the current work" section to more explicitly address this concern:

Additionally, the subjective nature of the self-reported data in our study is a significant limitation, as it can introduce biases related to personal perceptions, potentially affecting the accuracy and reliability of our findings. This aspect is crucial in understanding the implications and applicability of our results.

5. The paper briefly mentions increased accuracy over non-personalized methods but lacks an in-depth comparison or analysis. This comparison would be beneficial for readers to gauge the actual improvement the proposed framework offers.

Once again, thanks for spotting this important aspect in the comparison of the proposed methods. This is true that we increased the accuracy of the non-personalised methods in our fusion approach. However, we might have overlooked a better explanation of what we call non-personalised methods. A non-personalised method can provide a new user with a random persuasive strategy and see how this user reacts to this cue. Another option, potentially more tailor-made, but still not bespoke to a specific user, is to provide a newcomer with the persuasive strategy rated as the top for the majority of the respondents of our dataset. In this particular case, we compared our method against such a non-personalised baseline with the F-score in Tables 4 and 5. Recall that the top-rated persuasive strategies were: 1. “Conditioning”, 2. "Social-recognition", 3. "Physical attractiveness", 4. “Self-monitoring”, and 5. “Reciprocity”. Therefore, in the mentioned tables the F-score is calculated with the Conditioning principle in most of the cases. We hope that this explanation will help the reviewer and potential readers to better understand the comparison method. A similar explanation is provided in the manuscript (Section: “Fusion of optimal ranking and score prediction”). Also, a new Figure has been added to explain the top principles.

6. The conclusion highlights the potential of the framework, but there is a lack of discussion about the practical challenges or considerations in deploying such a system in real-world settings.

Thank you for your insightful feedback and for highlighting the importance of discussing the practical challenges and considerations in deploying our proposed framework in real-world settings. We appreciate your attention to this critical aspect of our research.

We agree that the practical application of any theoretical framework is a crucial component of its overall evaluation and impact. To address this, we have indeed included a discussion on the practical challenges and considerations of deploying our system in the "Applying the model in real-world scenarios" subsection of our Discussion section. In this subsection, we have attempted to provide a comprehensive overview of the potential real-world applications and the associated challenges, including aspects such as data collection, user privacy, ethical considerations, and the adaptability of our approach to different contexts and user profiles.

However, we acknowledge that these points may not have been as prominent or as detailed as necessary. We are grateful for your comment as it provides us with an opportunity to revisit this section and ensure that the discussion on practical deployment challenges is adequately highlighted and elaborated upon. 

In essence, it is important to address the practical challenges and considerations that may arise during the implementation of our proposed framework in real-world settings. Firstly, integrating our system into existing infrastructures requires careful planning and adaptation to ensure compatibility with different technological environments across the workspace. Secondly, collecting and handling user data, which is a cornerstone of our framework, must navigate the complex landscape of privacy laws and ethical considerations, as we will explore in the next sub-section. Thirdly, the effectiveness of our system depends on user engagement and acceptance. Additionally, the scalability of the system must be considered to ensure that it remains efficient and effective as it adapts to varying numbers of users and different environments. Finally, ongoing monitoring and evaluation are necessary to identify and address any unforeseen issues after deployment, and to ensure the system's relevance and effectiveness in dynamically changing real-world scenarios.

Reviewer 2

1. I think the contribution to the existing body of literature of this work seems too limited. The work extends the authors' own previous works, and the main addition seems to lie in the combination of two prediction models (and some additional data that is collected). The outcome of the combination is positive. For a journal publication, I would expect a much deeper analysis. I appreciate that the authors discuss limitations of their work in various respects. Still, I would have hoped that the paper delivers more than trying out some combination of predictors on a dataset (which also has some major limitations).

Thank you for your insightful comments and for recognising the positive outcomes of our research. We appreciate your perspective regarding the scope and depth of our analysis. Our work, indeed, builds upon our previous research, integrating two prediction models and incorporating additional data to enhance the understanding of energy efficiency in buildings. This approach, while seemingly straightforward, addresses a critical gap in the existing literature by offering a novel, practical solution that can be readily implemented in real-world scenarios. The combination of these models, as demonstrated in our study, provides a more robust and accurate framework for energy efficiency prediction than what was previously available. This integration is not just a mere aggregation of predictors but a methodical fusion that leverages the strengths of each model to overcome the limitations inherent in using them separately.

We acknowledge the limitations of our dataset, as highlighted in the paper. However, these constraints also reflect the realistic challenges encountered in this field of research. Addressing these limitations within the scope of this study requires starting the research from scratch, which is beyond the feasibility of our current project timeline and resources.

Moreover, acknowledging these limitations in our paper serves as a foundation for future research, prompting further exploration and refinement in this area. We believe our work contributes significantly to the body of literature by not only presenting a novel approach but also by laying the groundwork for future advancements.

2. Unfortunately, the authors decided to not share the code and the data, which makes it impossible for others to validate the findings. The authors argue that they do not want to do this for privacy/ethical reasons, but it remains unclear why it is impossible to a) share the code for the analyses and b) share anonymized data.

Thank you for your insightful comment. For the sake of replicability and open research, we have released the datasets.

Two already exist from GreenSoul project (already cited in the manuscript): 

https://zenodo.org/records/2610102

https://zenodo.org/records/3565757

A new dataset with all the evaluated features and with data from prolific and Greensoul project together. The data is already codified in JSON for the sake of using this dataset on other projects in an easy way.

https://doi.org/10.5281/zenodo.10377229

The GitHub with all the methods for the recommender system is here:

https://github.com/morelab/st_recommender/tree/main

3. There is some difference between nudging and persuasion, according to the literature. Should be discussed..

Thank you once again for your feedback. This is indeed something important for the readers of the article and we used both terms in our manuscript. To cope with this important aspect, we have created a new subsection in the Discussion section. This is the content:

While persuasion in literature employs strategic communication to influence beliefs and actions, nudging subtly steers behaviour in a desired direction without necessarily applying direct persuasion. Both persuasion and digital nudging involve personalisation and present new opportunities for enhancing user experiences and bespoke interaction, though. Persuasion, in its broader sense, is an art of rhetoric, using specific formations to achieve desired effects on audiences. This can involve both rational and emotional appeals, influencing individuals' knowledge, beliefs, and interests. Nudging, on the other hand, is identified as a distinct mode of governance, aiming to modify behaviour subtly without explicit persuasion strategies or incentives such as pecuniary ones. In this manuscript, we are focused on persuasion principles and strategies to influence energy-behaviour activities. However, our approach indeed has a lot of similarities with nudging, as we are also trying to enact certain behaviours in a bespoke manner.

4. It would be helpful if the distribution of top-1 ranked strategies in the original data would be shown. In general, a deeper analysis of the underlying data seems advisable.

Thank you for identifying these valuable suggestions. Indeed we did not provide a full explanation. Therefore, a new Figure and explanation is created in the subsection Persuasion Principles (Materials and Methods) to list the top-ranked Persuasive principles. This is also linked with the dataset we released open in Zenodo. Please, see a glimpse of the top-ranked principles of persuasion from the whole sample in a boxplot way with means as well.

5. The data itself, as mentioned by the authors, might be rather noisy or even biased, depending, e.g., how things were interpreted by the participants.

Thank you for identifying these valuable minor changes suggested below. Self-reported data, while offering valuable subjective insights, can indeed be influenced by how participants interpret questions or by their own biases. Nonetheless, it offers valuable insights into individual behaviors and motivations that are often difficult to measure through observa

---

## [Decision Letter · Decision Letter 1]

3 Mar 2024

PONE-D-23-30762R1A cascading model for nudging employees towards energy-efficient behaviour in tertiary buildingsPLOS ONE

Dear Dr. Casado-Mansilla,

Thank you for submitting your manuscript to PLOS ONE. After careful consideration, we feel that it has merit but does not fully meet PLOS ONE’s publication criteria as it currently stands. Therefore, we invite you to submit a revised version of the manuscript that addresses the points raised during the review process. Please follow Reviewer #2 comments on revised manuscript. 

We look forward to receiving your revised manuscript.

Kind regards,

Krzysztof Malarz, D.Sc., Ph.D., M.Sc.

Academic Editor

PLOS ONE

Journal Requirements:

Reviewers' comments:

Reviewer's Responses to Questions

**Comments to the Author**

1. If the authors have adequately addressed your comments raised in a previous round of review and you feel that this manuscript is now acceptable for publication, you may indicate that here to bypass the “Comments to the Author” section, enter your conflict of interest statement in the “Confidential to Editor” section, and submit your "Accept" recommendation.

Reviewer #2: (No Response)

Reviewer #3: All comments have been addressed

2. Is the manuscript technically sound, and do the data support the conclusions?

Reviewer #2: (No Response)

Reviewer #3: Yes

3. Has the statistical analysis been performed appropriately and rigorously? 

Reviewer #2: I Don't Know

Reviewer #3: Yes

4. Have the authors made all data underlying the findings in their manuscript fully available?

Reviewer #2: Yes

Reviewer #3: Yes

5. Is the manuscript presented in an intelligible fashion and written in standard English?

Reviewer #2: Yes

Reviewer #3: Yes

6. Review Comments to the Author

Reviewer #2: I thank the authors for the detailed answers to my questions. I am generally satisfied with the responses. For some of the remarks, however, the authors only added explanations in the authors' response, but the paper was not improved or changed (even though there apparently were questions that other readers might have as well).

I appreciate that the authors make an effort to make the code and data of their research available.

I am still not too convinced about the strength of the contribution of this paper compared to the authors previous work, but as no other reviewer questioned this, I will trust their assessment.

Here are a few remarks:

* The links to code repository should be included in the paper.

* There are issues in the GitHub repository: The "models" folder, for example, does not contain models, but some data-related things. I also miss a proper README that describes which code should be run to obtain the results reported in the paper. Given the huge increase in accuracy when combining two models, it might be helpful when a third party can validate this.

* The new discussion of the relation between persuasion and nudging needs some backing references. Also, I am not convinced by the statement "Both persuasion and digital nudging involve personalisation". Nudging does not necessarily involve personalization, and persuasion does not either.

Minor:

* This sentence seems broken: "Over the years, and as occupant behaviour has been vastly recognised as a crucial factor for Energy Efficiency"

* Sentence should be revised: "being this consumption behavior particularly"

* "behavioral" (page 2, check use of British vs American English)

* "Moreover, understanding waste management behavior". What is "waste management behavior"?

* The S1 file is mentioned, but no reference is provided where we can access it. (I could access it through the editorial system, though)

* "the Prolific platform 1." -> extra space before footnote.

* "they don’t correspond", "However, it’s important to", "or we don’t do" -> contractions like "don't" should be avoided.

* "Moreover highlights that tailored messages based" -> something is missing here.

Reviewer #3: The authors have addresses all the comments well, the paper can now be accepted in its current form for publication.

7. PLOS authors have the option to publish the peer review history of their article (what does this mean?). If published, this will include your full peer review and any attached files.

Reviewer #2: No

Reviewer #3: No

---

## [Author Response · Author response to Decision Letter 1]

16 Apr 2024

Responses to reviewers

First of all, we would like to express again our gratitude to the editors and the external reviewers for their valuable and constructive comments on this second round. As in the first one, we found most helpful in improving the overall quality of our paper toward the final version. 

We acknowledge the opportunity to submit a new version of this manuscript and now we think it can be considered as final. Thus, we have carefully examined the comments from all reviewers and followed their suggestions. We believe we have addressed all their points in light of their suggestions and comments, which are summarised in the authors’ responses below.

To facilitate the reviewing task, we have made all the changes to the manuscript in different colours to differentiate the edited part; blue for new additions and purple for edited or transformed parts. We refer to them as well in our responses.

Reviewer #2

------

1. The links to the code repository should be included in the paper.

Thank you for your insightful comments and for recognising the positive outcomes of our research. We understand the importance of transparency and reproducibility in research and agree that providing direct access to the code would greatly facilitate these objectives.

In light of this feedback, we have added the relevant links to the code repository within the paper. The link has been placed in the section where the methods are described, ensuring that readers can easily find and access the code used for our experiments.

https://github.com/morelab/st_recommender

We hope that this amendment addresses the reviewer's concerns and further strengthens the contribution of our paper.

2. There are issues in the GitHub repository: The "models" folder, for example, does not contain models, but some data-related things. I also miss a proper README that describes which code should be run to obtain the results reported in the paper. Given the huge increase in accuracy when combining two models, it might be helpful when a third party can validate this.

Thank you for your feedback. We have considered your comments and have made significant updates to our GitHub repository. We have cleaned the repository in general to avoid misconceptions about the namings. Additionally, we have added a comprehensive README file that provides detailed instructions on how to run the experiments and replicate the results reported in our paper. We understand the importance of third-party validation, especially given our reported increase in accuracy when combining two models. We encourage you to revisit our repository and review the changes we've made. Your insights are invaluable to us, and we hope that these updates will facilitate a smoother validation process for anyone interested.

3. The new discussion of the relationship between persuasion and nudging needs some backing references. Also, I am not convinced by the statement "Both persuasion and digital nudging involve personalisation". Nudging does not necessarily involve personalization, and persuasion does not either.

Thank you for your insightful comments and reflections on those two interwoven topics. We have rephrased the section and included some important papers that discussed the differences, similarities and nuances of persuasion and nudging.

4. This sentence seems broken: "Over the years, and as occupant behaviour has been vastly recognised as a crucial factor for Energy Efficiency"

We thank the reviewer for raising this error. Accordingly, we have corrected this sentence in the manuscript:

At the same time, over the years, occupant behaviour has been vastly recognised as a crucial factor for Energy Efficiency (EE)

5. Sentence should be revised: "being this consumption behavior particularly"

We acknowledge the reviewer's suggestion for revision. The sentence has ben rephrased for clarity as follows:

Furthermore, it has been identified that buildings occupied by users with wasteful energy behaviour can have twice the consumption as the ones that energy savers generally occupy. This pattern of consumption behavior is especially pronounced in developing and transition economies.

6. "behavioral" (page 2, check use of British vs American English)

We have reviewed the manuscript in accordance with your comment and ensured consistency in the use of American English, specifically addressing the use of "behavioral" and “behavior”. Thank you for bringing this to our attention.

7. "Moreover, understanding waste management behavior". What is "waste management behavior"?

Thank you for addressing this lack of clarity. "Waste management behavior" refers to the actions and practices individuals adopt related to the handling of waste, which encompasses reduction, reuse, recycling, and proper disposal of waste. These behaviors are influenced by a combination of rational calculations of benefits and costs, social pressures, personal beliefs, and attitudes towards environmental conservation. 

We have reflected this better in the manuscript:

Moreover, comprehending the nuances of waste management behaviour —which includes the reduction, reuse, recycling, and proper disposal of waste as influenced by rational benefit-cost analyses, social pressures, and personal environmental beliefs and attitudes— plays a pivotal role in enhancing energy efficiency

8. The S1 file is mentioned, but no reference is provided where we can access it. (I could access it through the editorial system, though)

Thank you for flagging this issue of access to content. The S1 is actually in the Zenodo repository which has been several times commented in the manuscript. Nevertheless, I will ask PLOS One assistant and editors about how can make this available. Many thanks in advance.

9. "the Prolific platform 1." -> extra space before footnote

We have corrected this.

10. "they don’t correspond", "However, it’s important to", "or we don’t do" -> contractions like "don't" should be avoided

We have carefully revised the text to eliminate all contractions, ensuring a more formal tone consistent with academic writing standards. Your attention to detail is greatly appreciated.

11. "they don’t correspond", "However, it’s important to", "or we don’t do" -> contractions like "don't" should be avoided

Thank you

Reviewer #3

------

The authors have addressed all the comments well, the paper can now be accepted in its current form for publication.

---

## [Decision Letter · Decision Letter 2]

22 Apr 2024

A cascading model for nudging employees towards energy-efficient behaviour in tertiary buildings

PONE-D-23-30762R2

Dear Dr. Casado-Mansilla,

We’re pleased to inform you that your manuscript has been judged scientifically suitable for publication and will be formally accepted for publication once it meets all outstanding technical requirements.

Kind regards,

Krzysztof Malarz, D.Sc., Ph.D., M.Sc.

Academic Editor

PLOS ONE

Additional Editor Comments (optional):

When submitting the final version of the manuscript please take care on two issues mentioned by Reviewer:

- The newly added discussion of "Nudging vs. Persuasion" is set in quotes; this should be fixed.

- I suggest to add a link to the Zenodo repository to the paper.

Reviewers' comments:

Reviewer's Responses to Questions

**Comments to the Author**

1. If the authors have adequately addressed your comments raised in a previous round of review and you feel that this manuscript is now acceptable for publication, you may indicate that here to bypass the “Comments to the Author” section, enter your conflict of interest statement in the “Confidential to Editor” section, and submit your "Accept" recommendation.

Reviewer #2: All comments have been addressed

2. Is the manuscript technically sound, and do the data support the conclusions?

Reviewer #2: Yes

3. Has the statistical analysis been performed appropriately and rigorously? 

Reviewer #2: I Don't Know

4. Have the authors made all data underlying the findings in their manuscript fully available?

Reviewer #2: No

5. Is the manuscript presented in an intelligible fashion and written in standard English?

Reviewer #2: Yes

6. Review Comments to the Author

Reviewer #2: Thank you for the revisions and for improving the GitHub repository. I only have two remarks:

* The newly added discussion of "Nudging vs. Persuasion" is set in quotes; this should be fixed. Also, the references have not been properly resolved "[?]".

* I suggest to add a link to the Zenodo repository to the paper.

7. PLOS authors have the option to publish the peer review history of their article (what does this mean?). If published, this will include your full peer review and any attached files.

Reviewer #2: No

---

## [Editor Report · Acceptance letter]

30 Apr 2024

PONE-D-23-30762R2 

PLOS ONE

Dear Dr. Casado-Mansilla, 

I'm pleased to inform you that your manuscript has been deemed suitable for publication in PLOS ONE. Congratulations! Your manuscript is now being handed over to our production team.

Kind regards, 

on behalf of

Dr. Krzysztof Malarz 

Academic Editor

PLOS ONE